# Novel class IIb microcins show activity against Gram-negative ESKAPE and plant pathogens

Benedikt M Mortzfeld[1,2]*, Shakti K Bhattarai[1,2], Vanni Bucci[1,2,3]

[1]Program in Microbiome Dynamics, University of Massachusetts Chan Medical School, Worcester, United States; [2]Department of Microbiology, University of Massachusetts Chan Medical School, Worcester, United States; [3]Program in Systems Biology, University of Massachusetts Chan Medical School, Worcester, United States

## eLife Assessment

This study presents **important** advances in the discovery and assessment of microcins that improve our understanding of their prevalence and roles. The bioinformatics analysis, expression, and antimicrobial assays are **solid**, although the diverging evaluations also indicated the need for additional support regarding the sequence analysis and validation to fully back some of the claims and conclusions. This study will appeal to researchers working on the discovery and analysis of novel peptide natural products.

*For correspondence:
benedikt.mortzfeld@umassmed.
edu

**Abstract** Interspecies interactions involving direct competition *via* bacteriocin production play a vital role in shaping ecological dynamics within microbial ecosystems. For instance, the ribosomally produced siderophore bacteriocins, known as class IIb microcins, affect the colonization of host-associated pathogenic *Enterobacteriaceae* species. Notably, to date, only five of these antimicrobials have been identified, all derived from specific *Escherichia coli* and *Klebsiella pneumoniae* strains. We hypothesized that class IIb microcin production extends beyond these specific compounds and organisms. With a customized informatics-driven approach, screening bacterial genomes in public databases with BLAST and manual curation, we have discovered 12 previously unknown class IIb microcins in seven additional *Enterobacteriaceae* species, encompassing phytopathogens and environmental isolates. We introduce three novel clades of microcins (MccW, MccX, and MccZ), while also identifying eight new variants of the five known class IIb microcins. To validate their antimicrobial potential, we heterologously expressed these microcins in *E. coli* and demonstrated efficacy against a variety of bacterial isolates, including plant pathogens from the genera *Brenneria*, *Gibbsiella*, and *Rahnella*. Two newly discovered microcins exhibit activity against Gram-negative ESKAPE pathogens, *i.e.*, *Acinetobacter baumannii* or *Pseudomonas aeruginosa*, providing the first evidence that class IIb microcins can target bacteria outside of the *Enterobacteriaceae* family. This study underscores that class IIb microcin genes are more prevalent in the microbial world than previously recognized and that synthetic hybrid microcins can be a viable tool to target clinically relevant drug-resistant pathogens. Our findings hold significant promise for the development of innovative engineered live biotherapeutic products tailored to combat these resilient bacteria.

## Introduction

A large body of theoretical and experimental work has shown that dynamics of microbiomes are shaped by the network of interbacterial interactions (*Stein et al., 2013*; *Bucci and Xavier, 2014*;

*Coyte et al., 2015*; *Hromada and Venturelli, 2023*). These cooperative and competitive interactions are often achieved *via* the secretion of cross-feeding metabolites (*Culp and Goodman, 2023*), antimicrobial peptides (*Heilbronner et al., 2021*), and bacterially produced small molecules (*Hibbing et al., 2010*; *Donia and Fischbach, 2015*) and are crucial to ecological properties including stability and ability to respond to external perturbations (*Coyte and Rakoff-Nahoum, 2019*; *Coyte et al., 2021*). Among the competitive interactions, bacteriocin production is proposed to be a prominent mediator of microbiome dynamics (*Niehus et al., 2021*) and, specifically, several reports including ours have shown that a bacteriocin subclass, class IIb microcins, mediates *Enterobacteriaceae* dynamics *in vivo* (*Sassone-Corsi et al., 2016*; *Mortzfeld et al., 2022*; *Cherrak et al., 2024*).

Class IIb microcins are ribosomally synthesized bacteriocins between 5 kDa and 10 kDa in size with activity against closely related strains or species (*Mortzfeld et al., 2022*; *de Lorenzo, 1984*; *Vassiliadis et al., 2010*; *Palmer et al., 2020*; *Baquero et al., 2019*). Unlike all other microcins, they carry a serine-rich C-terminal motif for a posttranslational modification with a siderophore, here an enterobactin or an enterobactin derivative, before they are secreted into the extracellular space (*Azpiroz et al., 2001*). Siderophores are iron-chelating molecules commonly employed by various bacteria to scavenge free iron to compete with other bacteria, particularly in resource-scarce environments such as the gastrointestinal tract (*Vassiliadis et al., 2010*; *Palmer et al., 2020*; *Patzer et al., 2003*) and are often associated with increased pathogenicity or virulence (*Miethke and Marahiel, 2007*; *Wilson et al., 2016*; *Khasheii et al., 2021*). The iron-chelating moiety of these posttranslationally modified antimicrobial peptides is recognized by high-affinity receptors and functions as a Trojan Horse key to susceptible bacteria as it triggers import into the periplasmic space, where the peptide inhibits the molecular target of susceptible bacteria (*Patzer et al., 2003*; *Bieler et al., 2006*; *Destoumieux-Garzón et al., 2003*; *Rodríguez and Laviña, 2003*). Because of these features, delivery of class IIb microcins by wildtype and engineered probiotics has been recently proposed as a strategy to combat drug-resistant enteric bacteria (*Sassone-Corsi et al., 2016*; *Mortzfeld et al., 2022*; *Palmer et al., 2020*; *Palmer et al., 2018*), which is in line with a growing body of work from the past decade that explores siderophore conjugation, including with enterobactin, to specifically deliver antibiotics and other small molecules to drug-resistant Gram-negative pathogens (*Page, 2019*; *Negash et al., 2019*; *Rayner et al., 2023*).

To date only five class IIb microcins have been described and only four have been characterized in terms of their antimicrobial activity. Specifically, the class IIb microcins MccE492 and MccG492 (uncharacterized) are solely present in *Klebsiella pneumoniae* (*Kp*), whereas MccH47 is specific for *Escherichia coli* (*Ec*) (*Vassiliadis et al., 2010*). Additionally, truncated versions of *mciA* (MccI47) and *mcmM* (MccM) are present in *Kp* RYC492, whereas they are intact in the *Ec* CA46 genome (*Vassiliadis et al., 2010*). Interestingly, while the genes encoding for microcin posttranslational modifications are highly conserved between *Ec* and *Kp*, suggesting a conserved pathway for microcin maturation, the toxin and corresponding immunity genes are significantly more variable (*Figure 1—figure supplement 1*). We hypothesized that class IIb microcin production extends beyond these specific compounds and organisms and identified a total of 12 novel class IIb microcins in seven additional *Enterobacteriaceae* species. Utilizing heterologous expression of these compounds in our *Ec* system optimized for enterobactin conjugation, we show potent antimicrobial activity by the encoded toxins against a library of bacteria, including Gram-negative ESKAPE and plant pathogens. This demonstrates that class IIb microcin genes are more prevalent in the microbial world than previously recognized and that synthetic hybrid microcins can be a viable tool to target clinically relevant drug-resistant pathogens.

## Results

With the hypothesis that class IIb microcin production is a common trait among *Enterobacteriaceae*, we anticipated that the genes encoding for the antimicrobial and immunity peptides would exhibit a high degree of dissimilarity to already known peptides as target specificity may result in accelerated adaptive coevolution (*Paterson et al., 2010*). Therefore, in addition to the known microcin and immunity genes from MccE492, MccG492, MccH47, MccI47, and MccM, we included in our informatic approach the genes necessary for mature class IIb microcin biosynthesis, extending our search to longer sequences for more reliable Basic Local Alignment Search Tool (BLAST) results (*Boratyn et al., 2012*). Moreover, we hypothesized that the amino acid sequences of genes responsible for posttranslational modification and microcin export would be less prone to evolutionary changes, thereby

maintaining the functional integrity of the gene cluster (*Vassiliadis et al., 2010*; *Lynch et al., 2016*). We then assessed their proximity in the respective genome location, because microcin genes are typically flanked by genes essential for toxin maturation (*Vassiliadis et al., 2010*). Further, we manually assessed and annotated small open reading frames (ORFs) upstream and downstream of the maturation genes, allowing us to also identify novel class IIb microcins without significant sequence similarity to the known antimicrobials, enabling the discovery of compounds with new molecular targets or modes of action (see Materials and methods).

Our informatics-driven analysis identified 12 promising class IIb microcin candidates from seven gene clusters with high similarity to *Ec* CA46 and *Kp* RYC492 in seven species across the *Enterobacteriaceae* family (*Figure 1A*, *Figure 1—figure supplement 2*): (i) *Brenneria goodwinii* (*Bg*; 2; GenBank: CP014137), (ii) *Gibbsiella quercinecans* (*Gq*; 1; CP014136), (iii) *Klebsiella oxytoca* (*Ko*; 1; CP033844), (iv) *Pantoea* sp. (*Ps*; 1; CP034363), (v) *Raoultella ornithinolytica* (*Ro*; 4; CP008886), (vi) *Salmonella enterica* (*Se*; 2; CP030220), (vii) *Serratia fonticola* (*Sf*; 1; CP033055). Although it has traditionally been a defining characteristic of class IIb microcins that all required genes are encoded within the chromosome (*Sassone-Corsi et al., 2016*), the gene cluster we discovered for *Se* is situated on a 159 kbp plasmid. Phylogenetic sequence analysis of both the antimicrobial and immunity peptide genes revealed the presence of eight different clades represented in both trees, respectively (*Figure 1B and C*). Regarding the well-established class IIb microcins MccH47, MccI47, MccM, MccG492, and MccE492, we identified novel members for each group, supported by nucleotide sequence similarity, amino acid identity, the closest *blastp* match, and domain predictions (*Table 1*, *Figure 1—figure supplement 3*). It is important to note that application of established tools for secondary metabolite identification (e.g. antiSMASH 7.0) (*Blin et al., 2023*) to these genomes did not yield identification of any of the old or novel microcins providing support of the relevance of our approach. In order to then ensure that these novel microcins are unique and not part of any other microcin class, we performed phylogenetic analysis for all known microcin genes from the classes I, IIa, and IIb and show distinct clustering for all newly described sequences (*Figure 1—figure supplement 4*). In light of this discovery, we propose a new nomenclature for class IIb microcins that includes the species initials in which they were identified (e.g. *Ec*, *Kp*), the closest relative already characterized class IIb microcin (G492, E492, H47, I47 or M), as well as the identifiers 'A' for antimicrobial or 'I' for immunity gene.

Based on this, the novel G492 relative found in *S. enterica* will be called *Se* G492 with the antimicrobial peptide identified as *Se* G492A and the immunity peptide identified as *Se* G492I. It is worth highlighting that in the case of the G492 group, all its members have the immunity gene located downstream of the antimicrobial gene, whereas for the other clades, this arrangement is reversed. In addition to uncovering eight novel variants of the five previously characterized microcins, we have identified four additional microcins through manual curation of ORFs in proximity to the microcin maturation genes. These novel microcins, which we name microcin W (MccW), microcin X (MccX), and microcin Z (MccZ), seem to belong to three entirely new clades based on nucleotide similarity (*Figure 1B and C*). The two members of the microcin X group, found in *B. goodwinii* (*Bg* X) and *R. ornithinolytica* (*Ro* X), only show significant similarity between one another, but not to any of the other antimicrobial or immunity peptides. This holds true for the nucleotide similarity (*Figure 1B and C*) as well as amino acid identity and the closest *blastp* hits (*Figure 1D*, *Table 1*, *Supplementary file 1*). Similarly, MccW from *G. quercinecans* (*Gq* W) does not show any sequence similarity to either the known or novel antimicrobial or immunity peptides in terms of nucleotide similarity, amino acid identity, the respective *blastp* hits, or phylogenetic localization (*Figure 1*, *Table 1*, *Supplementary file 1*). Lastly, MccZ from *R. ornithinolytica* (Ro Z) shows insignificant amino acid similarity with *Ec* MA (*mcmA*) for the antimicrobial, whereas the immunity peptide does not have any match among the known or the novel microcins (*Figure 1C*, *Table 1*, *Supplementary file 1*). Crucially, the identification of MccX, and MccZ within the same gene clusters as representatives of the E492 (*Bg* E492), H47 (*Ro* H47), and I47 (*Ro* I47) groups, strongly implies that they are functional components of a microcin gene cluster.

To test the newly identified microcins for antimicrobial activity, we used our previously established *Ec* overexpression system (*Mortzfeld et al., 2022*; *Palmer et al., 2020*). All antimicrobial and immunity peptides were codon optimized, synthesized, and cloned into an inducible high copy vector (see Materials and methods). Thus, we extracted the novel microcins out of their native genomic context of siderophore biosynthesis and transferred them into a heterologous expression background optimized for microcin-monoglycosylated enterobactin (MGE) linkage (*Mortzfeld et al., 2022*; *Palmer et al.,*

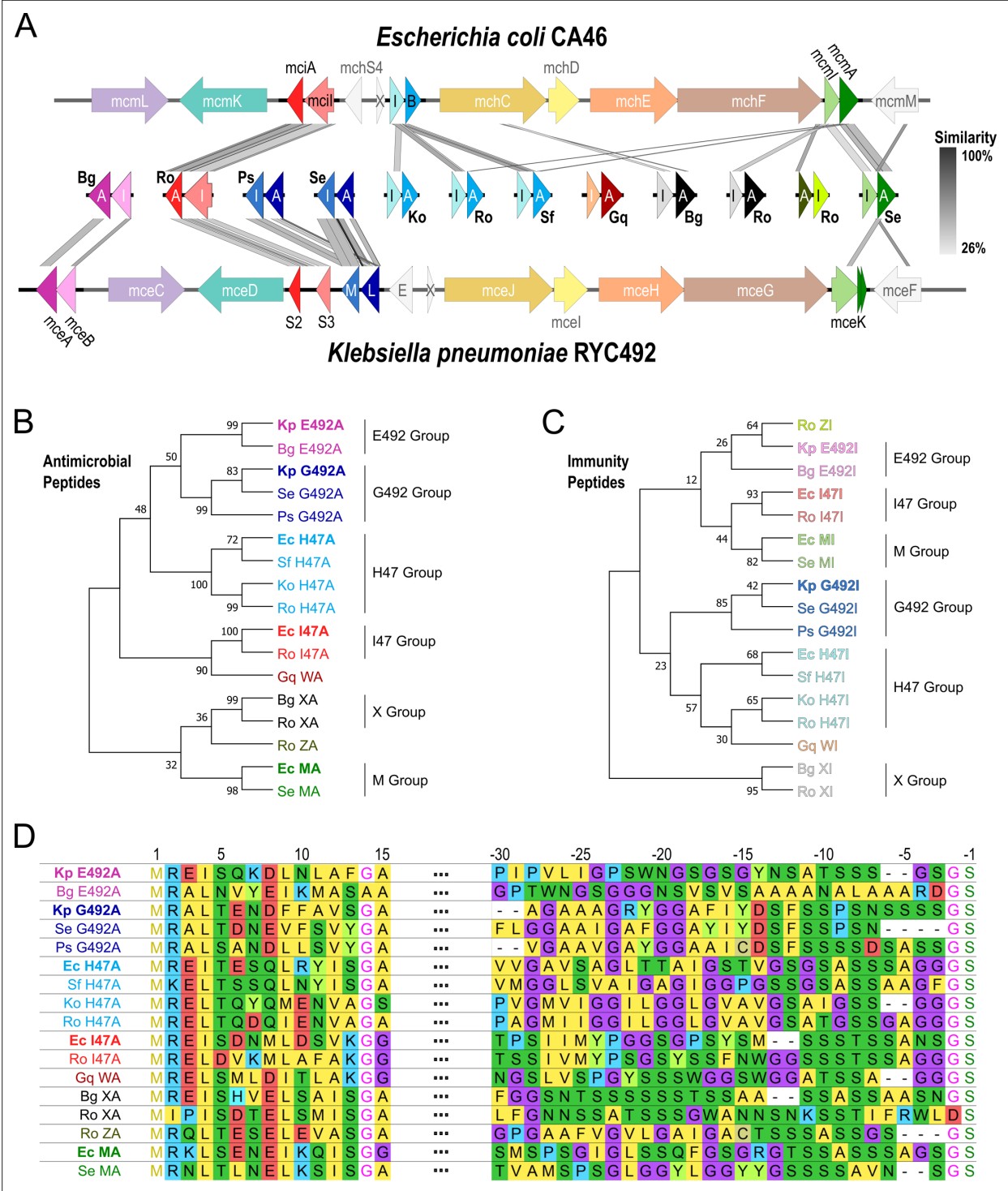

**Figure 1.** Novel class IIb microcins are found in numerous *Enterobacteriaceae* genomes. (**A**) Sequence alignments of the newly identified microcin and immunity genes with the gene clusters of *E. coli (Ec)* CA46 and *K. pneumoniae (Kp)* RYC492 using Easyfig (***Sullivan et al., 2011***). Antimicrobial (**A**) and immunity (**I**) genes in the center are represented by darker and lighter shades, respectively. X=mchX, I=mchI, B=mchB, E=mceE, L=mceL, M=mceM. (**B,C**) Phylogenetic trees of antimicrobial and corresponding immunity genes using codon-aligned nucleotide sequences with general time reversible model with discrete gamma distribution (GTR+G) and the Hasegawa-Kishino-Yano model with discrete gamma distribution (HKY+G), respectively. (**D**) MUSCLE (***Edgar, 2004***) alignment of the amino acid sequence of the signal peptide sequence as well as the C-terminus of the antimicrobial peptides.

The online version of this article includes the following figure supplement(s) for figure 1:

*Figure 1 continued on next page*

*Figure 1 continued*

**Figure supplement 1.** Comparison of class IIb microcin gene clusters from *E. coli* CA46 and *K. pneumoniae* RYC492.

**Figure supplement 2.** Comparison of class IIb microcin gene cluster from newly identified genomes with *E. coli* CA46 and *K. pneumoniae* RYC492.

**Figure supplement 3.** Region and domain prediction of antimicrobial (**A**) and immunity peptides (**B**) using SMART (*Letunic et al., 2021*).

**Figure supplement 4.** Phylogeny of all microcins from the classes I, IIa, and IIb.

*2020*). This allowed us to create hybrid compounds that could be efficiently tested for antimicrobial activity in an *E. coli* background. Through static plate inhibition assays involving live-producing cells (*Mortzfeld et al., 2022*; *Palmer et al., 2020*; *Palmer et al., 2018*), we successfully validated robust antimicrobial activity of 11 out of the 12 newly discovered microcins (*Figure 2A*). Notably, antimicrobial activity was only observed in iron-depleted media (*Figure 2B and C*). The hybrid microcins exhibit a range of specificities, with some inhibiting targets narrowly (e.g. *Ps* G492AI), while others exert a broader effect against multiple bacteria (e.g. *Se* G492AI). Moreover, our study also provides the first evidence of inhibitory activity by *Kp* G492, a microcin whose existence and function have only been proposed in the scientific literature based on genetic sequence (*Vassiliadis et al., 2010*).

To this date class IIb microcins have been only shown to be very selective and only active against different species within the *Enterobacteriaceae* family (*Mortzfeld et al., 2022*; *Vassiliadis et al., 2010*; *Palmer et al., 2020*; *Palmer et al., 2018*). While the activity for the novel microcins varies, we here report, for the first time, antimicrobial activity outside of the *Enterobacteriaceae* family utilizing hybrid antimicrobial peptides. We demonstrate that microcins *Ps* G492 and *Se* G492 have activity against Gram-negative multidrug-resistant ESKAPE pathogens with both being capable of inhibiting *A. baumannii* (BAA 1790), and with microcin *Se* G492 alone also showing activity against *P. aeruginosa* (PA14) (*Figure 2*). Specifically, compared to *K. pneumoniae* (BAA 1705) *Se* G492 is 256 times more effective against *A. baumannii* (BAA 1790), 128 times more effective against *E. coli* (BAA 196), and 8 times more effective against *P. aeruginosa* (PA14) (*Figure 2B*).

## Discussion

With a comprehensive analysis of publicly available bacterial genomes, we unraveled 12 previously undiscovered class IIb microcins. Among these findings, we identified three novel microcin clades,

**Table 1.** Blastp results and closest matches to the known class IIb microcins MccE492, MccG492, MccH47, MccI47, or MccM. Red color indicates no significant match found.

| Microcin name | Species | Accession no. | Antimicrobial gene name | Antimicrobial closest match | Identical/total length | E-value | Immunity gene name | Immunity closest match | Identical/total length | E-value |
|---|---|---|---|---|---|---|---|---|---|---|
| *Bg* E492 | *Brenneria goodwinii* | CP014137 | *Bg* E492A | *Kp* E492A (mceA) | 35/106 | 2.00E-14 | *Bg* E492I | *Kp* E492I (mceB) | 14/94 | 6.00E-04 |
| *Bg* X | *Brenneria goodwinii* | CP014137 | *Bg* XA | *Kp* E492A (mceA) | 21/98 | 6.90E-02 | *Bg* XI | *Ec* MI (mcmI) | 10/78 | 2.60E+00 |
| *Gq* W | *Gibbsiella quercinecans* | CP014136 | *Gq* WA | *Ec* I47A (mciA) | 7/103 | 8.10E+00 | *Gq* WI | *Ec* MI (mcmI) | 8/74 | 3.30E+02 |
| *Ko* H47 | *Klebsiella oxytoxa* | CP033844 | *Ko* H47A | *Ec* H47A (mchB) | 32/77 | 1.00E-07 | *Ko* H47I | *Ec* H47I (mchI) | 19/69 | 2.00E-14 |
| *Ps* G492 | *Pantoea* sp. | CP034363 | *Ps* G492A | *Kp* G492A (mceL) | 43/89 | 6.00E-09 | *Ps* G492I | *Kp* G492I (mceM) | 37/85 | 2.00E-20 |
| *Ro* Z | *Raoultella ornithinolytica* | CP008886 | *Ro* ZA | *Ec* MA (mcmA) | 21/64 | 1.00E-03 | *Ro* ZI | *Ec* MI (mcmI) | 5/74 | 2.00E+01 |
| *Ro* H47 | *Raoultella ornithinolytica* | CP008886 | Ro H47A | *Ec* H47A (mchB) | 31/79 | 2.00E-07 | Ro H47I | *Ec* H47I (mchI) | 21/69 | 4.00E-14 |
| *Ro* I47 | *Raoultella ornithinolytica* | CP008886 | *Ro* I47A | *Ec* I47A (mciA) | 37/79 | 4.00E-26 | Ro I47I | *Ec* I47I (mciI) | 64/135 | 1.00E-43 |
| *Ro* X | *Raoultella ornithinolytica* | CP008886 | *Ro* XA | *Ec* MA (mcmA) | 18/99 | 4.40E-01 | Ro XI | *Ec* MI (mcmI) | 9/61 | 1.00E-02 |
| *Sf* H47 | *Serratia fonticola* | CP033055 | *Sf* H47A | *Ec* H47A (mchB) | 38/87 | 8.00E-11 | *Sf* H47I | *Ec* H47I (mchI) | 48/68 | 4.00E-22 |
| *Se* G492 | *Salmonella enterica* | CP030220 | *Se* G492A | *Kp* G492A (mceL) | 56/85 | 1.00E-11 | *Se* G492I | *Kp* G492I (mceM) | 38/84 | 1.00E-23 |
| *Se* M | *Salmonella enterica* | CP030220 | *Se* MA | *Ec* MA (mcmA) | 34/90 | 1.00E-17 | *Se* MI | *Ec* MI (mcmI) | 38/71 | 6.00E-04 |

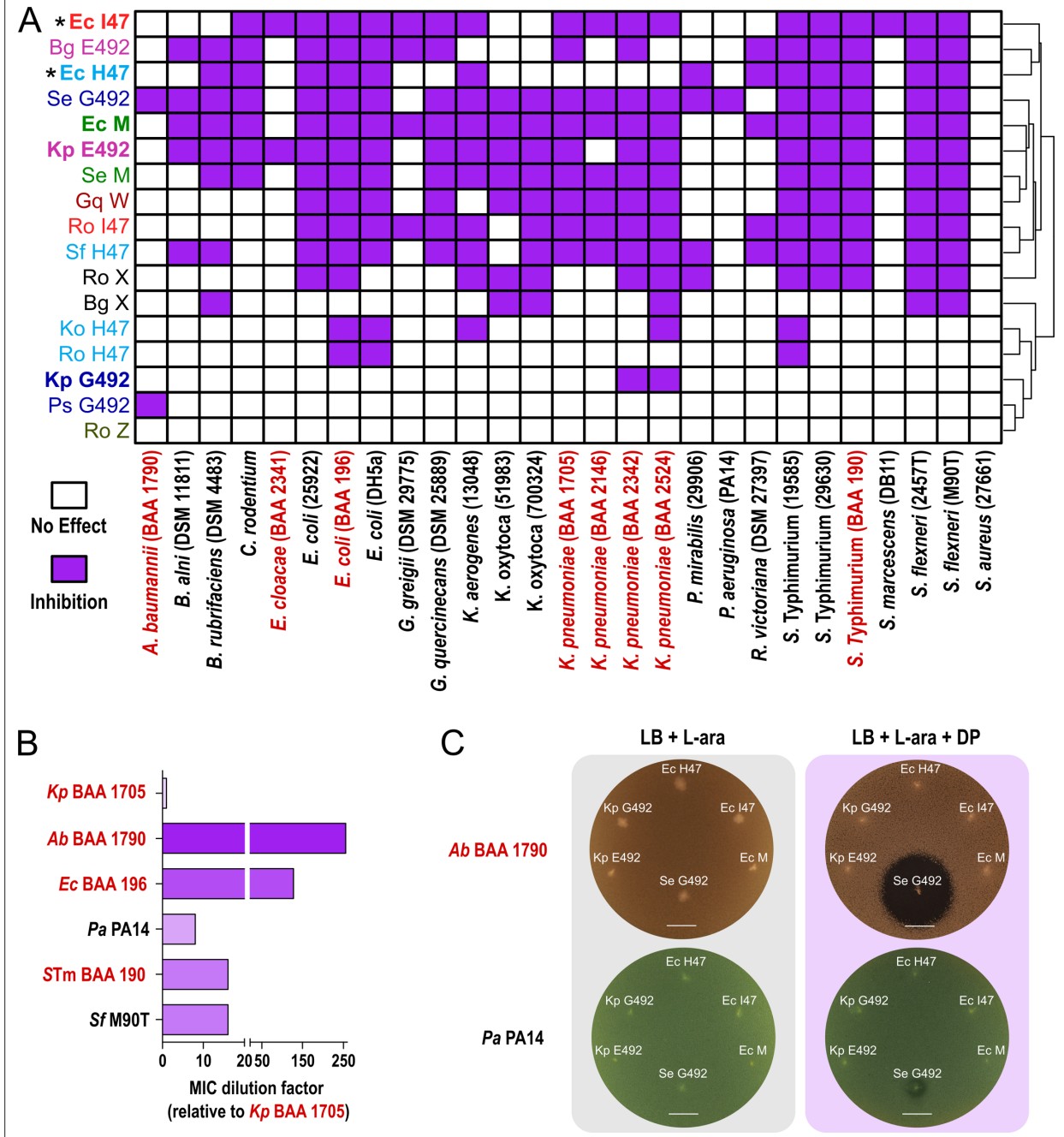

**Figure 2.** Novel class IIb microcins are effective inhibitors of *Enterobacteriaceae* and Gram-negative ESKAPE pathogens. (**A**) Heatmap summarizing the inhibitory potential of known and novel class IIb microcins against a library of *Enterobacteriaceae*, *Pseudomonadales*, and Gram-positive bacteria, including multidrug-resistant isolates (red) as determined by static inhibition assays with live-producing bacteria. *=activity determined through microcin purification and minimum inhibitory concentration assays (*Mortzfeld et al., 2022*; *Palmer et al., 2020*). (**B**) Relative minimum inhibitory concentrations for *S. enterica* (*Se*) G492 against different bacterial species. Note that *Se* G492 is 256 times more potent against *A. baumannii* (BAA 1790) compared to *K. pneumoniae* (BAA 1705). (**C**) Static inhibition assays comparing *K. pneumoniae* (*Kp*) E492, *Kp* G492, *Ec* H47, *E. coli* (*Ec*) I47, *Ec* M, and *Se* G492 activity from single colony production against multidrug-resistant *A. baumannii* (BAA 1790) and *Pseudomonas aeruginosa* (PA14). Note that iron-limited conditions (DP) are required for antimicrobial activity, confirming action of class IIb microcins. L-ara=L-arabinose, DP = 2,2-dipyridyl, scale bars: 1 cm.

specifically MccW, MccX, and MccZ. Through heterologous expression, we showed antimicrobial activity for all but one novel microcins and are the first to demonstrate activity for the known class IIb microcin *Kp* G492. Hence, this research demonstrates that class IIb microcin genes exhibit a higher prevalence in *Enterobacteriaceae* genomes than previously reported. As a result, their impact on

ecological community dynamics in natural environments, including the growth of *Pseudomonadales* species, might be broader than previously thought. For antimicrobial activity testing, microcin and immunity genes were overexpressed recombinantly in our *Ec*-derived expression system optimized for microcin-MGE production (*Mortzfeld et al., 2022*; *Palmer et al., 2020*). The common process of posttranslational modification with the siderophore consolidated the import mechanism of the hybrid microcins toward enterobactin, the most characteristic siderophore of the Enterobacteriaceae family. This allowed us to test the target-specific antimicrobial activity of the microcins irrespective of siderophore production in the native genomic background. However, it is important to note that class IIb microcin activity is dependent on active import through siderophore receptors and consequently some of these microcins might display different activity spectrums when tested in their native genomic background of siderophore biosynthesis. Furthermore, static plate inhibition assays exhibit lower sensitivity compared to purification approaches with quantitative minimum inhibitory concentration (MIC) assays. Thus, the activity spectrums of the hybrid microcins could encompass a wider range than what has been described in this study when tested as a purified product. However, historically the microcin literature proves that ideal approaches for purification and MIC testing can vary between the antimicrobials (*Mortzfeld et al., 2022*; *Vassiliadis et al., 2010*; *Palmer et al., 2020*; *Destoumieux-Garzón et al., 2003*).

We were able to expand the origins of class IIb microcins from the enteric bacteria *Ec* and *Kp* to other members of the *Enterobacteriaceae* family, including well-known phytopathogens (*Denman et al., 2012*; *Brady et al., 2010*; *Allahverdipour et al., 2020*). Specifically, *B. goodwinii* and *G. quercinecans* are associated with acute oak decline and are frequently isolated together (*Denman et al., 2018*) and the two strains containing microcin genes were isolated within the same research project. Notably, these bacteria grow synergistically (*Brady et al., 2022*), while upregulating iron transporters during co-culture (*Doonan et al., 2020*), hinting at class IIb microcin-related competition. We were able to show activity of the overexpressed hybrid microcins against human-derived enteric isolates, however, their native spectrum might have evolved to target more frequently encountered strains from the genus *Brenneria* or *Gibbsiella*. Further, we demonstrated activity of several class IIb microcins against the three tree pathogen genera *Brenneria*, *Gibbsiella*, as well as *Rahnella* (*Brady et al., 2022*; *Brady et al., 2014*; *Hajialigol et al., 2023*; *Moradi-Amirabad and Khodakaramian, 2020*; *Poret-Peterson et al., 2019*). Thus, treatment with potent microcins, purified or produced in live bacteria, could present a viable option to target bacteria-caused plant diseases.

In health care settings the burden by Gram-negative ESKAPE pathogens and multidrug-resistant *Enterobacteriaceae* weighs heavily on modern medicine and novel antimicrobials are needed to develop new treatment options (*WHO, 2014*; *World Health Organization, 2017*). In addition to enteric pathogens and pathobionts, bacteria outside of the *Enterobacteriaceae* family have also been shown to scavenge for and to import enterobactin, including *P. aeruginosa* and *A. baumannii* (*Moynié et al., 2019*; *Subashchandrabose et al., 2016*). Therefore, different siderophore conjugates could be a viable option to target these pathogens as well or to fine-tune the desired target range (*Page, 2019*; *Negash et al., 2019*; *Rayner et al., 2023*). Antimicrobial peptides and particularly microcins are promising candidates for selective eradication of enteric pathogens and have been demonstrated to potently reduce pathogen colonization *in vivo*, when produced by a live probiotic (*Sassone-Corsi et al., 2016*; *Mortzfeld et al., 2022*). Here, we present the most comprehensive library of class IIb microcins created so far, that is suited for heterologous expression and *in vivo* application for the development of novel live biotherapeutic products against drug-resistant enteric bacteria and Gram-negative ESKAPE pathogens.

In this study, we challenge the prevailing notion that class IIb microcin production is limited to *Ec* and *Kp*. Through comprehensive genomic analysis of publicly available bacterial genomes, coupled with heterologous overexpression, we unveiled a set of undiscovered class IIb microcins across *Enterobacteriaceae* species. Our findings not only expand the known repertoire of class IIb microcins but also hold significant implications for synthetic hybrid compounds. We demonstrate that these newly identified class IIb microcins exert remarkable inhibitory effects on ESKAPE pathogen species when expressed in a system for enterobactin-derived conjugation. This discovery underscores their potential as agents against a broader spectrum of pathogens, including those affecting humans and plants, thus opening new avenues for antimicrobial research and applications.

## Materials and methods

### Bioinformatic class IIb microcin identification

We developed a pipeline that by leveraging BLAST (*Boratyn et al., 2012*) enabled us to mine publicly available genome databases for novel, previously undescribed class IIb microcins. We included *mchCDEF* and *mcmL* for *Ec* as well as *mceCDGHIJ* for *Kp* for posttranslational modification and export, expecting more reliable hits for longer and functionally conserved proteins in close proximity to class IIb microcin and immunity genes. Thus, we first ran *tblastn* (*Boratyn et al., 2012*) against RefSeq (*O'Leary et al., 2016*), to screen for all genes related to biosynthesis pathways, known microcin genes, as well as immunity gene sequences exhibiting homology to the microcin gene clusters found in *Ec* CA46 and *Kp* RYC492. Homology to the microcin gene clusters were guided by BLAST parameters sseqid (genome ID), pident (percentage of identical positions) along with sstart (start of alignment in genome) and send (end of alignment position in genome). Resulting hits were concatenated by genome ID and assessed for their proximity to one another in the genome. These gene clusters should, at best, contain all the known genes required for toxin maturation, including *mchCDEF* and *mcmL* (*Vassiliadis et al., 2010*; *Palmer et al., 2020*). In addition to genomic hits to the known microcins, small ORFs of 50–150 amino acids in size close to the biosynthesis genes were screened and annotated manually using the criteria described below as well as their domains were predicted using SMART (*Letunic et al., 2021*). The ORFs were meticulously examined and assessed against established class IIb microcin criteria known from *Ec* H47, *Ec* I47, *Ec* M, *Kp* E492, and *Kp* G492: (i) a serine-rich C-terminus culminating in a final serine, (ii) the presence of fewer than two cysteine residues, (iii) a signal peptide within the initial 15 amino acids ending with GG or GA, and (iv) close proximity (≤200 bp) to an ORF featuring a predicted transmembrane domain, typically encoding an immunity peptide. The identified genes were included in the pipeline's input to expand the scope of gene detection. We repeated this process iteratively through the pipeline until no additional genes were added to the output. Subsequently, *blastp* was used to assess microcin similarity as shown in *Table 1* and *Supplementary file 1*.

### Phylogenetic analyses

For the native full-length coding sequence of the microcin and immunity genes, a codon-based sequence alignment was generated using the MUSCLE algorithm (*Edgar, 2004*). For phylogeny of all microcins, the nucleotide sequences without the respective signal peptides were codon-aligned. Subsequently, we determined the best fit substitution models for maximum likelihood phylogenetic analyses, resulting in the general time reversible model with discrete gamma distribution (GTR+G) and the Hasegawa-Kishino-Yano model with discrete gamma distribution (HKY+G), respectively. A bootstrap test with 1000 replicates for maximum likelihood and random seed was conducted for all trees. Alignment, model testing, and tree building were performed in MEGA11 (*Tamura et al., 2021*).

### antiSMASH analyses

To test if similar results of class IIb microcin identification could be obtained with automated bioinformatic tools, we ran antiSMASH 7.0 (*Blin et al., 2023*), a widely used tool for microbial genome mining and biosynthetic gene cluster detection. As input, we utilized the seven genomes from the newly identified class IIb microcins: (i) *Bg* CP014137, (ii) *Gq* CP014136, (iii) *Ko* CP033844, (iv) *Ps* CP034363, (v) *Ro* CP008886, (vi) *Se* CP030220, (vii) *Sf* CP033055. As a positive control for the well-established microcins *Kp* E492 and *Kp* G492 as well as *Ec* H47 and *Ec* M, we used the accession numbers CP127839 (*Kp* RYC492) and CP148105 (*Ec* Nissle 1917), respectively. Notably, using the 'loose' setting, in none of the cases a class IIb microcin biosynthesis gene cluster was detected, nor were any microcin genes identified. This was the case for both, the novel microcins and the original, well-annotated, microcins.

### Plasmids and heterologous class IIb microcin expression

ORFs of identified microcin and immunity genes were codon optimized for frequent *Ec* codon usage without creating repetitive sequences and synthesized by Integrated DNA Technologies (Coralville, IA, USA) with 18 bp of native 5′ upstream sequence and 20 bp of native 3′ downstream sequence, respectively. Using Gibson Assembly (*Gibson et al., 2009*), the genes were cloned into our previously established *Ec* class IIb microcin expression system that results in mature class IIb microcins posttranslationally modified with an MGE (*Mortzfeld et al., 2022*; *Palmer et al., 2020*). Briefly, the

antimicrobial and the immunity genes are co-expressed under the control of an arabinose-inducible pBad/araC promoter in a high copy plasmid with a pUC-derived origin of replication. All assemblies were verified using whole plasmid sequencing. DNA files for all used plasmids can be found as supplementary material.

## Static inhibition assays

Cultures of strains with confirmed plasmid assemblies were spread on LB agar plates containing 100 µg/ml ampicillin. In addition to a pUC19 control without microcin expression, single colonies for each microcin were picked with a sterile pipet tip and all stabbed into the same solid LB agar plate containing 100 µg/ml ampicillin for plasmid retention, 0.2 mM 2,2-dipyridyl to create iron-limited conditions during the growth phase, and 0.4% L-arabinose for induction of gene expression. Plates were incubated at 37°C for up to 72 hr, before they were overlaid with the target bacterial isolates. Note that testing all microcin-expressing stains on the same plate allowed us to confidently assess differential inhibitory activity between all 17 tested microcins. For the overlay, the microcin-producing bacteria in the stabs were inactivated using chloroform vapors and 10 min under ultraviolet light. Then, target bacteria were diluted 1:2000 from overnight culture in LB media containing 100 µg/ml ampicillin and 0.2 mM 2,2-dipyridyl. *Ec* and *Shigella flexneri* strains were diluted 1:200 to acquire dense bacterial lawns. Finally, 0.5 ml of molten agar was added to 2 ml of liquid media and the resulting soft agar was spread on the plate with the inactivated bacteria and incubated for 16 hr at 37°C. The pUC19 control strain was unable to create any zone of inhibition against any of the tested target bacteria. Note that not all tested target bacteria confer resistance to ampicillin, however, placing stabs for 17 ampicillin-resistant microcin producers and the negative control evenly spaced into a single plate degrades the antibiotic within 24 hr of incubation and thus before the overlay with the target bacteria is added.

## Relative MIC dilution factors

For enrichment of microcin *Se* G492, an MBP-microcin fusion protein was expressed from pHMT-SeG492 in *E. coli* BL21 cells as previously described (*Mortzfeld et al., 2022*; *Palmer et al., 2020*). Cells harvested from 6 l culture were resuspended in 50 ml column buffer (200 mM NaCl, 20 mM Tris-HCl, pH 7.5), lysed by sonication, and passed through 5 ml of high flow amylose resin (New England Biolabs, Ipswich, MA, USA) as recommended by the manufacturer. The protein was eluted with 30 ml 10 mM maltose, cleaved with Tobacco etch virus (TEV) protease, and further processed as previously reported (*Mortzfeld et al., 2022*; *Palmer et al., 2020*). After removal of the histidine-tagged TEV protease, the relative MIC assays were conducted using sterile 96-well round bottom microplates. The plates were prepared as follows: the first row contained 20 µl of 2× LB with 0.4 mM 2,2'-dipyridyl and 20 µl of *Se* G492 containing solution in amylose resin elution buffer (200 mM NaCl, 20 mM Tris-HCl, 10 mM maltose, pH 7.5). All other wells were filled with 20 µl of 1× LB, 0.2 mM 2,2'-dipyridyl, and 0.5× amylose resin elution buffer, and a twofold serial dilution was performed across the plate. The target bacteria were grown overnight in LB at 200 rpm and 37°C and were added to a final dilution of $10^{-4}$ into the wells. The plates were then incubated in the dark at 37°C with gentle agitation. Relative MICs were determined as the lowest concentration at which no growth was observed after 24 hr. All reported values represent the median of at least three biological replicates.

## Acknowledgements

The authors would like to thank the administrative staff in the Microbiology Department at the University of Massachusetts Chan Medical School, especially Annette Bohigian, Amy Parker, Dhruti Desai, Marie Berardi, Richard Fish, and Tracey Rae, for their support. This work was supported by the CDMRP PRMP W81XWH2020013 to VB, by the NIH NIA 1R01AG075283-01A1 to VB, and by the Deutsche Forschungsgemeinschaft (DFG) project 457837076 to BMM. The funders had no role in study design, data collection, and analysis, decision to publish, or preparation of the manuscript.

# Additional information

## Competing interests

Vanni Bucci: receives support from a sponsored research agreement from Vedanta Biosciences, Inc. The other authors declare that no competing interests exist.

## Funding

| Funder | Grant reference number | Author |
| --- | --- | --- |
| Congressionally Directed Medical Research Programs | W81XWH2020013 | Vanni Bucci |
| National Institutes of Health | 1R01AG075283-01A1 | Vanni Bucci |
| Deutsche Forschungsgemeinschaft | 457837076 | Benedikt M Mortzfeld |

The funders had no role in study design, data collection and interpretation, or the decision to submit the work for publication.

## Author contributions

Benedikt M Mortzfeld, Conceptualization, Data curation, Formal analysis, Supervision, Funding acquisition, Validation, Investigation, Visualization, Methodology, Writing – original draft, Project administration, Writing – review and editing; Shakti K Bhattarai, Conceptualization, Resources, Data curation, Software, Formal analysis, Investigation, Methodology, Writing – review and editing; Vanni Bucci, Conceptualization, Funding acquisition, Writing – original draft, Project administration, Writing – review and editing

## Author ORCIDs

Benedikt M Mortzfeld (iD) https://orcid.org/0000-0003-0563-8970
Shakti K Bhattarai (iD) http://orcid.org/0000-0001-9118-9184
Vanni Bucci (iD) https://orcid.org/0000-0002-3257-2922

Reviewer #1 (Public review): https://doi.org/10.7554/eLife.102912.2.sa1
Reviewer #2 (Public review): https://doi.org/10.7554/eLife.102912.2.sa2
Reviewer #3 (Public review): https://doi.org/10.7554/eLife.102912.2.sa3
Author response https://doi.org/10.7554/eLife.102912.2.sa4

# Additional files

## Supplementary files

• Supplementary file 1. Blastp results and closest matches to the known or novel class IIb microcins. Red color indicates no significant match found.

• MDAR checklist

## Data availability

The genomes are accessible with the following GenBank numbers: *Brenneria goodwinii* (CP014137), *Gibbsiella quercinecans* (CP014136), *Klebsiella oxytoca* (CP033844), *Pantoea sp.* (CP034363), *Raoultella ornithinolytica* (CP008886), *Salmonella enterica* (CP030220), *Serratia fonticola* (CP033055). All information is included in the manuscript or supporting files. All plasmid sequences as well as annotation files to produce Figure 1A, Figure 1-figure supplement 1, Figure 1-figure supplement 2, and Figure 1-figure supplement 3 are available as supplementary material.

The following previously published datasets were used:

| Author(s) | Year | Dataset title | Dataset URL | Database and Identifier |
|---|---|---|---|---|
| Minogue T, Wolcott M, Wasieloski L, Aguilar W, Moore D, Jaissle J, Tallon LJ, Sadzewicz L, Zhao X, Vavikolanu K, Mehta A, Aluvathingal J, Nadendla S, Yan Y, Sichtig H | 2018 | *Klebsiella oxytoca* strain FDAARGOS_500 chromosome, complete genome | https://www.ncbi.nlm.nih.gov/nuccore/CP033844.1 | NCBI GenBank, CP033844.1 |
| Zhou J, Xia F, Che S, Wang J, Qiu L, Li G, Shao J, Zhang G, Zhong L, Liu Q, Ren B | 2018 | *Pantoea sp.* CCBC3-3-1 chromosome, complete genome | https://www.ncbi.nlm.nih.gov/nuccore/CP034363.1 | NCBI GenBank, CP034363.1 |
| Leung F, Liu L, Jiang J | 2014 | *Raoultella ornithinolytica* strain A14, complete sequence | https://www.ncbi.nlm.nih.gov/nuccore/CP008886.1 | NCBI GenBank, CP008886.1 |
| Nash JHE, Robertson J, Bessonov K | 2018 | *Salmonella enterica* strain SA20021456 plasmid pSA20021456.1, complete sequence | https://www.ncbi.nlm.nih.gov/nuccore/CP030220.1 | NCBI GenBank, CP030220.1 |
| Bekkelund AK | 2018 | *Serratia sp.* 3ACOL1 chromosome, complete genome | https://www.ncbi.nlm.nih.gov/nuccore/CP033055.1 | NCBI GenBank, CP033055.1 |
| Doonan J, Denman S, McDonald JE | 2016 | *Brenneria goodwinii* strain FRB141, complete genome | https://www.ncbi.nlm.nih.gov/nuccore/CP014137.1 | NCBI GenBank, CP014137.1 |
| Doonan J, Denman S, McDonald JE | 2016 | *Gibbsiella quercinecans* strain FRB97, complete genome | https://www.ncbi.nlm.nih.gov/nuccore/CP014136.1 | NCBI GenBank, CP014136.1 |

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
