## [Editor Report · eLife Assessment]

This study presents **important** advances in the discovery and assessment of microcins that improve our understanding of their prevalence and roles. The bioinformatics analysis, expression, and antimicrobial assays are **solid**, although the diverging evaluations also indicated the need for additional support regarding the sequence analysis and validation to fully back some of the claims and conclusions. This study will appeal to researchers working on the discovery and analysis of novel peptide natural products.

---

## [Referee Report · Reviewer #1 (Public review)]

Summary:

*Enterobacteriaceae* produce microcins to target their competitors. Using informatics approaches, the authors identified 12 new microcins. They expressed them in *E. coli*, demonstrating that the microcins have antimicrobial activity against other microbes, including plant pathogens and the ESKAPE pathogens Pseudomonas aeruginosa and Acinetobacter baumannii.

Strengths:

Overall, this study has the merit of identifying new potential antimicrobial molecules that could be used to target important pathogens. The bioinformatics analysis, the expression system used, and the antimicrobial assays performed are solid, and the data presented are convincing. This work will set the basis for new studies to investigate the potential role of these microcins in vivo.

Weaknesses:

The work has been performed in vitro, which is a valid approach for identifying the antimicrobial peptides and assessing their antimicrobial activity. Future studies will need to address whether these new microcins exhibit antimicrobial activity in vivo (e.g., in the context of infection models), and to identify the targets (receptor and mechanisms of action) for the new microcins.

---

## [Referee Report · Reviewer #2 (Public review)]

Mortzfeld et al. describe their study of class IIb microcins. Furthering our awareness of the presence and action of microcins is an important line of research. However, several issues related to the premise, sequence analysis, and validation require attention to support the claims.

(1) Previous studies have been published on the broader distribution of microcins across bacteria. The software has been published for their identification. Comparison to this software and/or discussion of previous work should be included to place this work in the context of the field.

(2) It is not clear how immunity proteins were identified and there does not appear to be functional confirmation to show these predicted immunity proteins are real. Thus, it is premature to state that immunity genes have been found. This may also confound some of the validation studies below if proper immunity proteins have not been included.

(3) Please show the nt alignment used to generate the tree. Without seeing it, one would guess that the sequences are either quite similar (making the results from this study less novel) or there would be concerns that the phylogenetic relationship derived from the nt alignment is spurious.

(4) Figure 1 B-C: There are numerous branches that do not have phylogenetic support (values <50%). These are not statistically valid phylogenetic relationships and should be collapsed. The resulting tree should be used in the description of clades.

(5) The discovered microcins are not being directly tested since they are expressed heterologous and reliant on non-native modification systems. The results present the statement that novel microcins have been validated. This should be described accordingly.

(6) The key finding of this paper is the claim that 12 novel class IIb microcins have been validated. To substantiate this claim, original images showing evidence of antibacterial activity must be made available rather than a presence/absence chart. The negative controls for this table are unclear and should be included with the original images.

(7) Further data for the purified microcin is needed. The purification method described is standard practice and should allow for product quantification, which should be included. Standard practice includes an SDS page showing the purity of the microcin, or at least the TEV digest to show microcin has been produced, and importantly a control sample (scrambled sequence, empty vector purification, etc) to show that observed activity (Figure 2B) is not from a purification carry over. This data should be included to support that microcin has been purified and is active.

---

## [Referee Report · Reviewer #3 (Public review)]

Summary:

In this study, several novel class IIb microcin biosynthetic gene clusters have been discovered by specific homology searches and manual curation. Using a specific *E. coli* expression system, the microcins were expressed and conjugated to monoglycosylated enterobactin as siderophore moiety. While this synthetic biology approach cannot account for other siderophores being coupled to the microcin core peptide in the original producing strains, it nonetheless allows for a general screening for the activity of the heterologously produced compounds. Through this approach, the activity of several predicted microcins has been confirmed and three novel class IIb microcin clades were identified.

Strengths:

The experimental design is sound, the results are corroborated by suitable controls, and the findings have a high level of novelty and significance. Furthermore, the comments of the initial round of peer review have been answered satisfactorily by the authors.

---

## [Author Response]

We thank the anonymous reviewers very much for dedicating their time to thoroughly review our manuscript. We sincerely appreciate their thoughtful consideration and detailed assessment. Regarding the raised concerns, we acknowledge the importance of exploring the full scope of class IIb microcins, however, we believe that in depth characterization, purification, and *in vivo* application of the 12 novel compounds goes beyond the scope of this short report and discovery article.

At the same time, the reviewers acknowledge that the analysis, experimental design, the expression system as well as the performed assays are “sound”, “convincing”, and “corroborated by suitable controls”. In the present manuscript we sought to identify novel antimicrobials and to comprehensively verify their antimicrobial activity in *E. coli* irrespective of the siderophore-dependent delivery mechanism. Notably, none of the reviewers questioned that we describe new antimicrobials, the characteristics we used to find them, that they are class IIb microcins, or that they do exhibit antimicrobial activity against Gram-negative ESKAPE and plant pathogens.

We believe that our discovery study can serve as a steppingstone towards the application of bacterially produced antimicrobial compounds to target Gram-negative pathogens in numerous plant and animal species, including humans.